# Using the consolidated framework for implementation research to evaluate a model of community-engaged research in advance care planning

Erika VanDyke[1]*, William Calo[1], Benjamin Levi[1], Amy Tucci[2], Lauren Jodi Van Scoy[1], the Project Talk Trial workgroup[¶]

1 Penn State College of Medicine, 500 University Drive, Hershey, Pennsylvania, United States of America,
2 Hospice Foundation of America, Washington DC, United States of America

¶ The complete membership of the Project Talk Trial workgroup authors can be found in the Acknowledgments.
* erikanvandyke@gmail.com

## Abstract

### Background

Advance care planning (ACP) is the process of discussing one's goals and wishes for end-of-life care with loved ones or clinicians and then completing an advance directive (AD). Our Community-Based Delivery Model (CBDM) has demonstrated success in engaging these communities, yet the implementation mechanisms behind its effectiveness remain unclear. This study utilized the Consolidated Framework for Implementation Research (CFIR) to evaluate the CBDM in the context of the Project Talk Trial (PTT), a national randomized controlled trial of ACP interventions.

### Methods

This study employed a two-pronged approach. First, CFIR was used to systematically map the CBDM, defining domains and constructs relevant to the intervention's implementation in diverse community contexts. Second, semi-structured interviews with 24 community hosts who facilitated PTT events provided qualitative insights into the "inner setting," "outer setting," and "implementation process" domains. Deductive coding and thematic analysis were used to identify key implementation strategies and challenges.

### Results

The CFIR mapping revealed three critical features driving the CBDM's success: the transfer of resources between outer and inner settings, the central role of community hosts in bridging these domains, and the flexibility to adapt to local contexts. Semi-structured interviews identified five themes, including hosts' use of relational

**Data availability statement:** The de-identified interview transcripts supporting this study are publicly available in the Inter-university Consortium for Political and Social Research (ICPSR) data repository. Data can be accessed at: https://doi.org/10.3886/E246722V1.

**Funding:** This study is funded by the National Institutes of Health (R01MD014141). The funder had no role in the design of the study, data collection, data analysis, interpretation of data, or writing the manuscript.

**Competing interests:** The authors declare that the research was conducted in the absence of any commercial or financial relationships that could be construed as a potential conflict of interest. Dr. Van Scoy is an unpaid scientific advisor for Common Practice, LLC.

connections, teaming and engaging strategies, and culturally tailored approaches, which facilitated implementation. Notably, rural hosts exceeded recruitment goals, challenging the notion that rural populations are "hard to reach".

## Conclusions

This approach provides actionable insights for improving ACP efforts in communities settings. The integration of CFIR mapping and empirical data highlights the CBDM's potential as a scalable model for implementing community-engaged health interventions.

## Background

Advance care planning (ACP) is the process of discussing one's goals and wishes for end-of-life care with loved ones or clinicians and then completing an advance directive (AD) [1]. ACP provides significant benefits for patients, their families, and healthcare systems by helping to ensure that end-of-life care aligns with individual preferences. A recent systematic review of multiple randomized controlled trials has associated ACP interventions with improved patient-centered outcomes including patient-physician communication, reduced decisional conflict, and better patient-caregiver congruence in end-of-life care preferences [2]. Other well-designed randomized controlled trials have demonstrated that ACP reduces the likelihood that family members experience long-lasting stress, depression, and anxiety after surrogate decision-making [3].

Despite these known benefits, ACP rates in the US remain consistently below 40% [4–11]. Barriers to ACP completion include issues related to healthcare mistrust, stigmas about discussing death and dying, and reluctance to engage in end-of-life conversations [12–18]. These barriers translate to lower-quality end-of-life care, as individuals who have not performed ACP are more likely to receive care that is not concordant with their stated preferences [19–22], are less likely to receive hospice services [23–25], and are 3 times more likely to die in an intensive care unit while receiving burdensome and unwanted treatments that are unlikely to improve outcomes [19]. To address these barriers, fostering trust and establishing effective community outreach mechanisms are essential for increasing ACP participation.

Traditional ACP interventions are often delivered within the healthcare system [26,27], however, these approaches often fail to address healthcare mistrust and other barriers that limit ACP engagement. Community-based interventions, outside of a clinical environment and facilitated by trusted local leaders, may be more effective in engaging individuals in ACP discussions.

As such, we developed the community-based delivery model (CBDM) to bring ACP interventions to communities by leveraging existing and well-established community networks [28]. The CBDM involves collaboration with trusted community "hosts" (who are members of local organizations) to deliver a community-based activity in familiar and trusted venues (e.g., places of worship, senior centers, community centers) [28].

These venues are typically located outside the healthcare system in an effort to offset participants' discomfort and skepticism. In our model, hosts apply via an online application, are interviewed, and strategically chosen based on their experience organizing community events in their community and their understanding of best practices around community- engagement. To minimize administrative and marketing burdens on local organizations, hosts are provided access to an online portal that contains all the necessary training and event materials. Hosts are invited to include their organization's logo on project materials and then leverage their community networks to advertise the event in familiar settings. For events where research is conducted, research assistants travel to the sites to obtain informed consent and collect research data. To bolster acceptance and research participation within the community, the hosts facilitate the introduction of the research team and the event and research protocols to help attendees decide whether they wish to participant in the activity, the research, or both. A recent national study utilizing the CBDM demonstrated its potential, with 1,222 individuals attending ACP-related events at 53 community sites [28]. Remarkably, 90% of attendees consented to participate in research. Even more impressive, 98% of attendees completed at least 1 ACP behavior. This high rate of ACP behavior highlights the CBDM's effectiveness in engaging community groups in ACP engagement.

Building on the recognized limitations of ACP interventions within healthcare settings, the CBDM has demonstrated potential in enhancing participant recruitment and engagement by leveraging community-based strategies. However, there are still critical knowledge gaps that impede its broader implementation and sustainability. Specifically, we lack fundamental understanding of the strategies and processes that make hosts successful and the implementation obstacles that may present challenges. Furthermore, no systematic examination exists of hosts' operational processes or the barriers they face during implementation. Such information could help optimize the CBDM's components and develop practical guidance and support systems needed to expand this promising model while maintaining its effectiveness across varied community contexts.

To address these gaps, the present study leveraged the Consolidated Framework for Implementation Research (CFIR) [29] to systematically evaluate the model's implementation and explore host experiences within the Project Talk Trial (PTT), a national randomized control trial of community-based advance care planning (ACP) interventions. The present study had two objectives: (1) systematically map the implementation of the CBDM using the CFIR, and (2) qualitatively explore its implementation within the PTT. The data presented here evaluate the CBDM and examine host experiences, including implementation processes and barriers. These findings aim to inform researchers and community organizations seeking to enhance the implementation of the CBDM and improve ACP among underserved populations.

## Methods

### Guiding framework: CFIR

For this study, the CFIR was selected for its robust, adaptable framework, which allows for evaluating contextual factors in real-world settings.

As a deterministic framework, the goal of the CFIR is to explain barriers and facilitators of implementation effectiveness.[25] Thus, by utilizing the CFIR we assessed differences in implementation strategies of hosts via determinants across implementation settings. The CFIR is comprised of five domains (and corresponding constructs) that give researchers and public health practitioners the ability to better understand the implementation of an intervention. The "innovation" domain refers to the "thing" being implemented, which in the case of PTT refers to the ACP interventions (further described below). The "inner setting" domain is the location and context where the intervention is being implemented (i.e., the community setting of the research activity). The "outer setting" is where the "inner setting" can be found (i.e., the state a community event is held). The "implementation process" domain refers to the activities and strategies used to implement the intervention. Lastly, the "individuals" domain refers to those who receive the intervention and those who provide the intervention and/or involved with its implementation.[25] Our study focused largely on mapping the trial to this framework and understanding the host experience within the "inner setting" (which we defined as their community, networks, and

partners), the relationship with the "outer setting" (if any exist), and the "implementation process" (how they prepared, planned, and implemented the event).

## The Project Talk Trial

The PTT is a three-armed, cluster, randomized control trial comparing the efficacy of two ACP interventions, the *Hello* game and *The Conversation Project Starter Guide*, that use conversation-based approaches to promote ACP in diverse, underserved populations (placebo/attention control is T*able Topics*) [30]. The *Hello* game is a well-studied community intervention that has demonstrated efficacy in promoting ACP behavior change [28,31–35]. The game is comprised of a booklet of 32 questions, where in groups of 4–6 people, participant s share and respond to questions related to end-of-life preferences and medical decision making. *The Conversation Project Starter Guide* [36] is a nationally recommended tool to promote ACP (theconversationproject.org). The program includes a didactic presentation and facilitates group discussion related to ACP. The attention control arm involves a table-top conversation game called *Table Topics* that does not involve ACP conversations. Randomization into the arms occurs at the host level (described below) and is stratified by race/ethnicity and whether the site is considered urban or rural (as classified by the Rural Health Information Hub "Am I Rural?" tool). At the community event, attendees who provide informed consent participate in research data collection. The primary outcome is completion of an AD six-months after the event. Additional outcomes include engagement in other ACP behaviors (Fig. 1). The trial, which is ongoing, is registered at clinicaltrials.gov [NCT04612738] and is IRB-approved through Penn State (STUDY00014689).

## PTT community host participants

Community hosts (e.g., hospice organizations, hospitals, faith organizations, senior centers, and other community-based organizations) are recruited by the Hospice Foundation of America (HFA) via the organization's social media platforms and extensive national email list-serv. Interested hosts complete an online application and are interviewed by the research team. Individual hosts -- typically community outreach educators/specialists, nurses, social workers,or faith leaders -- who work on behalf of the community hosts demonstrate a history of successful community engagement are offered an

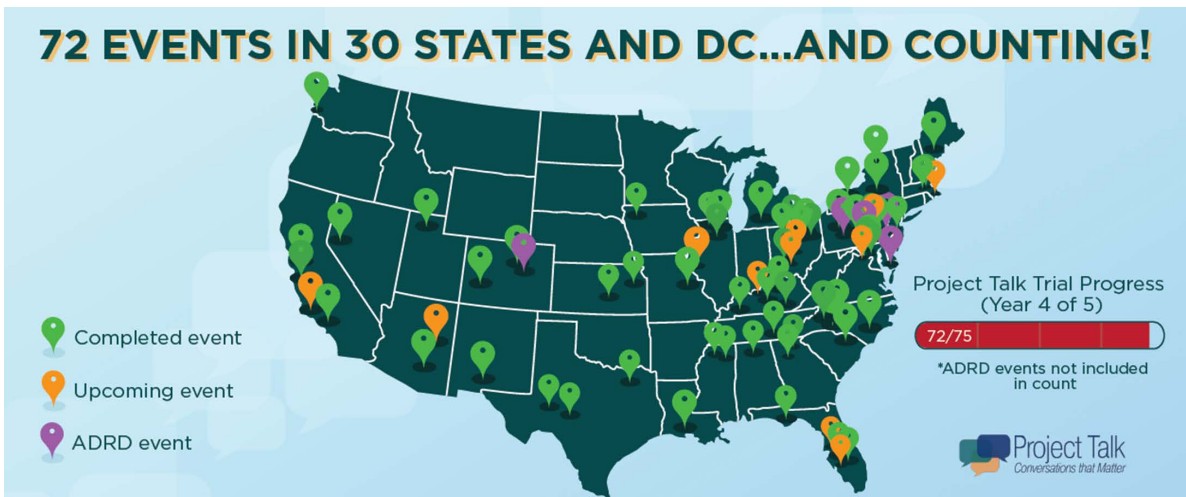

**Fig 1. Progress of the Project Talk Trial through November 29, 2023 including randomization strata.** Green pins indicated events that have already been completed; orange pins indicate events that are scheduled but not yet held, and purple pins indicate sites enrolled in a supplemental study involving participants with mild cognitive impairment or Alzheimer's Disease and Related Dementias (ADRD).

opportunity to participate in the study. Enrolled hosts are provided access to an online resource portal, where they complete three hours of both intervention-specific and research-related training. Through the portal, hosts can also access recruitment materials (e.g., flyers, press releases, social media templates) to advertise their event. After completing the training, hosts are randomized to one of the three interventions. Hosts are encouraged to hold events with at least 20 participants (for all arms). Hosts are provided a $300 stipend to assist with event related costs, including space rental and refreshments. Additional details regarding the trial protocol and CBDM have been published previously [28,30].

## CFIR mapping process

To better understand how the features of the CBDM function within the context of the PTT and align with the CFIR [29], we followed a process which involved defining each CFIR domain and its relevant constructs using project-specific language to systematically map the implementation of the PTT's CBDM. To validate and refine this mapping, two authors reviewed transcripts from host interviews (see below) to confirm the alignment of the mapped constructs with observed implementation processes and ensure a thorough understanding of how hosts interacted with both the project and the CBDM.

## Qualitative interviews

To explore the implementation of the CBDM in the PTT from the host perspective, we conducted qualitative interviews with the first 24 host participants. PTT hosts were interviewed 2–4 weeks after their community event using a semi-structured interview guide, which was developed using CFIR domains including "implementation process," "inner setting," "outer setting," and "innovation." These constructs were selected to best understand the activities and behaviors hosts engage in to plan the event and how their planning is actualized within their community. Interview questions that explored the "implementation process" included, "*Tell me how you went about organizing your event?*" and "*How did you work or collaborate with your partner to advertise the event, if at all?*" To better understand the "inner setting," hosts were asked, "*Did you partner with anyone outside of your organization to arrange the event, such as church or town leaders? If so, what role did these individuals play?*" To explore the "innovation," the interview guide asked questions such as, "*What were your thoughts about the host portal?*" and "*How did this event compare to tother ACP-related outreach events you've hosted?*" Interviewers EV, PM, KF, and HC reviewed the consent form with partcipants, and participants verbally consented prior to the virtual interview (as approved by Penn State IRB). Verbal consent was documented internally in REDCap. Permission to record was obtained by all participants. Audio files were transcribed verbatim using a professional transcription service. The first 24 host interviews of the PTT were included in this analysis, at which point saturation was reached with regard to interviewee qualitative data. Interviews were conducted between July 18, 2022 and June 12, 2024.

## Qualitative analyses

A thematic analysis was conducted on transcribed host interviews conducted between March 2022 and October 2024. Two analysts coded host interviews (EV and HB). The analysts utilized a deductive codebook comprised of the 5 domains and 48 constructs of the CFIR model. MAXQDA (v. 24) was used to organize and code the data.[37] Five transcripts (20% of the dataset) were coded by both analysts. After discussing discrepancies and adjusting coded segments, Cohen's kappa coefficient of 0.81 confirmed high inter-rater reliability. Analyst EV coded 13 transcripts, while analyst HB coded the remaining 6 transcripts. Themes related to the CFIR model and the implementation of the CBDM are included in these analyses. Consolidated Criteria for Reporting Qualitative Research (COREQ) guidelines for rigor were followed in the reporting of these results. Qualitative data were stratified and analyzed by comparing hosts with differing recruitment rates (<20 participants vs. > 20 participants) and those who held events in rural versus urban areas. Additionally, data were examined across the PTT strata (Urban-Black, Urban-White, Rural-Black, Rural-White, and Spanish-speaking sites).

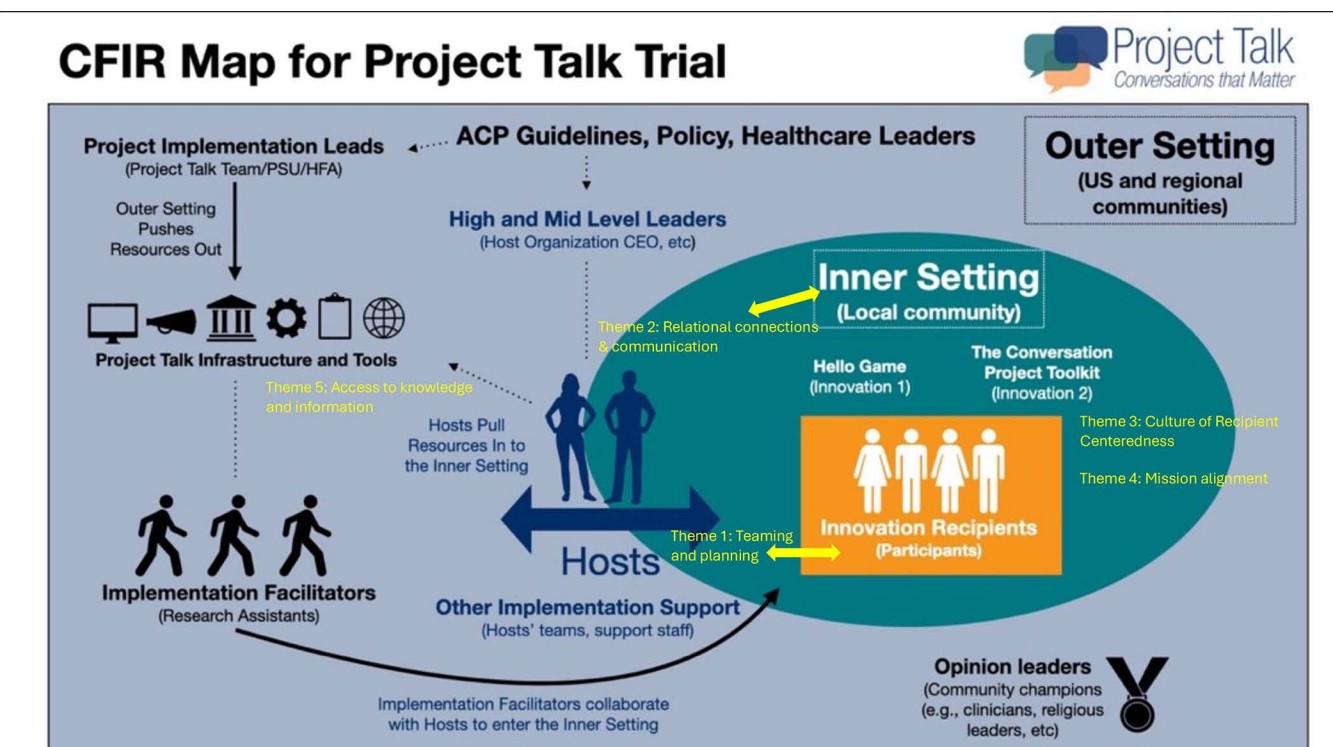

## Results

### Mapping the CFIR: Inner and outer settings

**Outer setting definition.** A key aspect of implementation involves understanding the settings where the intervention, in this case, ACP interventions, occurs. Within the CFIR, the outer setting encompasses the broader environment in which the inner setting is embedded. For PTT, the outer setting was defined broadly to include the 4 geographic regions of the US (Northeast, South, Midwest, West) where host organizations were recruited. It also included the healthcare organizations operating within these regions, the Project Talk team, and the associated project infrastructure. Federal legislation, policies, and healthcare guidelines advocating for ACP were also considered part of the outer setting.

Heterogeneity was observed across the outer setting, with variations in state laws and cultural attitudes toward end-of-life issues and ACP. For instance, differences in legal requirements for ADs, such as notarization, necessitated adaptations in language and approach. Regional and cultural differences also influenced perceptions and engagement with ACP. Additionally, the diversity of Spanish dialects across US communities posed challenges to intervention delivery and required linguistic modifications to ensure inclusivity and clarity.

**Inner setting definition.** The inner setting for this trial was defined as the local community and the specific venues where ACP interventions took place, such as churches, senior centers, and other community gathering spaces. Within these venues, the intervention was delivered directly to participants, making the inner setting a critical focal point for understanding implementation (Fig 2).

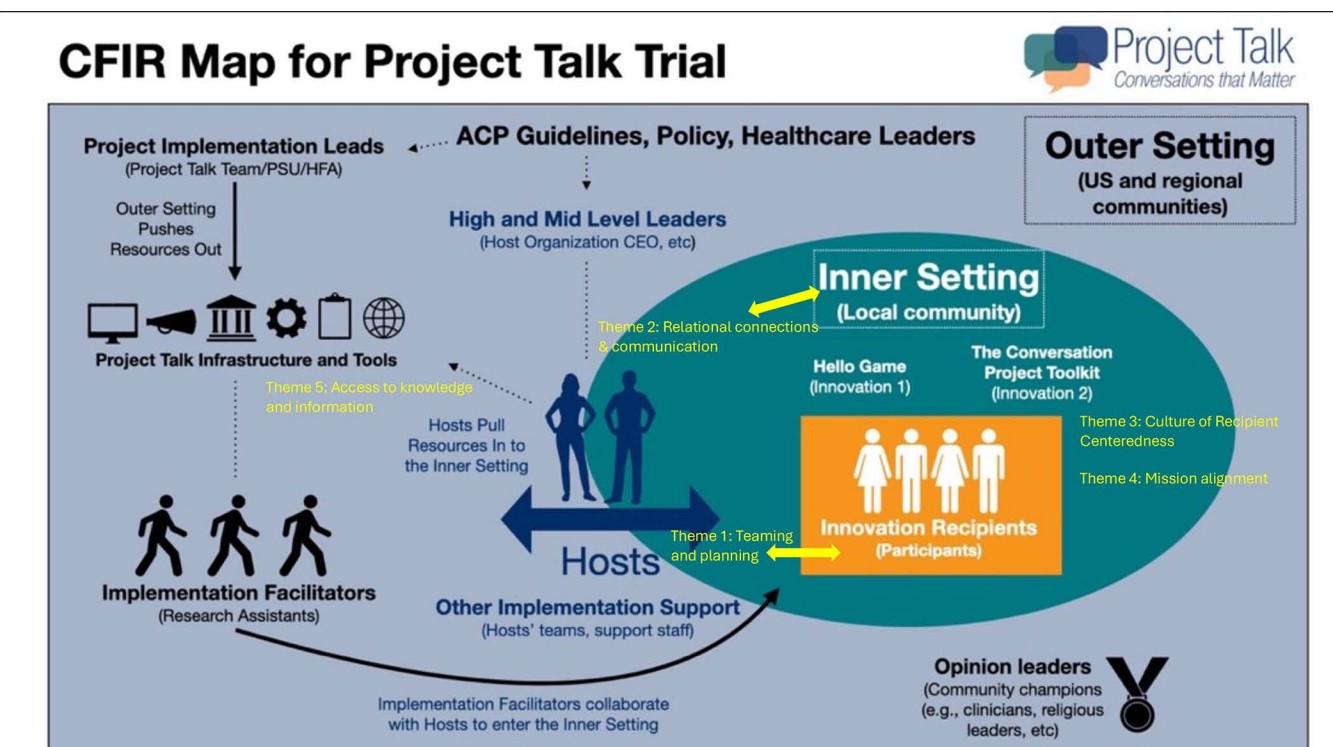

**Fig 2. Mapping based on the CFIR include elements of the outer setting (blue), the inner setting (teal) where the individual participants receive the innovation (orange).** The hosts are located centrally, spanning both the inner and outer setting. Blue text denotes transitions from the outer setting to the inner setting.

**Key structural elements.** Fig 2 provides a visual representation of how the CBDM connects outer and inner settings, highlighting the role of hosts in bridging these domains. Community hosts—serving as implementation leads—were central to this process. These hosts, affiliated with diverse organizations such as local hospice agencies, hospitals, academic institutions, and non-profits, brought resources and infrastructure from the outer setting into the inner setting.

### Qualitative interview study results

**Host characteristics.** Of the 26 hosts interviewed, 10 (38%) were affiliated with a hospice organization and seven (27%) with a medical center. Almost 54% of community hosts held an event at a community center, with other hosts held events at a religious organization (34%) or restaurant (4%). Hosts from a total of 18 states were included in the sample from across the Northeast (19%), Midwest (15%), South (42%) and West (23%). Hosts in this sample predominately held events in urban Black communities (39%) and Spanish-speaking communities (30.8%). See Table 1 for additional host characteristics.

**Thematic analysis.** Themes and corresponding CFIR domains and constructs are reported in Table 2 and Fig 2. In total, five themes were identified; one related to the "implementation process" domain and four related to the "inner setting" domain.

**Theme 1: Hosts described "teaming" and "engaging" strategies when planning their community event during the "implementation process." Those who used multiple "engaging" strategies to plan events experienced better recruitment outcomes.** After completing their required online training, hosts engaged in "teaming" behaviors, or joining

**Table 1. Host Characteristics.**

| Characteristics | No (%) |
| --- | --- |
| **Hosts (n = 26)** | |
| ***Host Event Strata*** | |
| Urban Black | 10 (38.4%) |
| Rural Black | 2 (7.7%) |
| Urban White | 3 (11.5%) |
| Rural White | 3 (11.5%) |
| Spanish-Speaking | 8 (30.8%) |
| ***Organization type*** | |
| Medical center | 7 (26.9%) |
| Hospice organizations | 10 (38.4%) |
| Healthcare organization (non-profit) | 6 (23.1%) |
| Academic Institution | 2 (7.7%) |
| Place of Worship | 1 (3.8%) |
| ***Venue types*** | |
| Community centers | 14 (53.8%) |
| Religious organizations | 9 (34.6%) |
| Medical Center | 2 (7.7.%) |
| Restaurant | 1 (3.8%) |
| ***# Events Held in Each Region*** | |
| Northeast | 5 (19.2%) |
| Midwest | 4 (15.4%) |
| South | 11 (42.3%) |
| West | 6 (23.2%) |
| *States represented* | 18 |

**Table 2. Themes and CFIR Frequencies.**

| Themes | CFIR construct |
|---|---|
| Theme 1: Hosts engaged in several strategies that varied in complexity to plan events, secure community venues, and target outreach through "teaming" and "engaging" which ultimately impacted their recruitment target. | Teaming<br>Planning |
| Theme 2: Within the "inner setting" hosts leverage their existing community "relational connections" to plan and host events | Relational connections<br>Communications |
| Theme 3: Within the "inner setting", Spanish-speaking event hosts used intentional, community-engaged approaches in their planning behaviors to host an event that would be well received by the community, demonstrating a "culture of recipient centeredness". | Culture of recipient centeredness |
| Theme 4: Within the "inner setting", hosts are willing to plan community research events because the event demonstrates "mission alignment" with the host's organization to promote advance care planning. | Mission alignment |
| Theme 5: Within the "inner setting," "access to knowledge and information" provided by the host training portal was critical to preparing for the research event in their community. | Access to knowledge and info |

together as a group, to intentionally coordinate and collaborate on tasks to host the research intervention. After delineating roles and responsibilities, the first task for many hosts was to find a venue for the event. Most hosts expressed little to no difficulty in securing a venue to host the event.

> "Once we got the okay that we were approved… I had reached out to a friend of mine who was a pastor at a local church to see if they would be interested in participating with us and providing a venue. His church is very active in the community and they have a social justice initiative. So, this seemed to be a perfect tie in. And so, he was very much on board." – Midwestern Host

Many hosts were able to host the event at their organization, as cited by this host:

> "We hosted it at our hospice building actually. So, there wasn't a lot of logistical stuff that needed to be done, which was nice." – Northeastern Host

After securing the venue, hosts transitioned to "engagement" activities, or methods to attract and encourage participation in the intervention. Hosts who used a multi-strategy approach to secure community participants (i.e., a combination of radio advertisements, flyers, and other advertising) typically met or exceeded their recruitment target of 20 participants. High recruiting hosts also conducted individual outreach (i.e., talking 1:1 with community members), which was often reported by hosts who held events in predominately Black communities. This host expressed how they personally invited the participants:

> "I think we were supposed to contact five people a piece…. I invited a lot of people myself. That's how we had such a great number on that weekend." – Southern Host

Notably, all rural hosts exceeded recruitment goals, averaging 25 participants per event, compared to the sample average of 22 participants per event. This rural host describes their multi-strategy to recruiting participants for their community research event:

*"…we did a lot of outreach through our social media pages. And through our webpage. This is a small town in rural [state]. So, a big part of how we get information out is through our local radio station.... We also advertised with… a commodities program through our office… to try to get people signed up that way, as well.* – Southern Host

Hosts who did not meet the recruitment target of 20 described utilizing recruitment methods that were more passive nature, with the majority of low-recruiters relying on just one strategy. One such host describes relying on a single community leader to conduct outreach for the event:

*"We met with one particular pastor and his team about the event, and we talked about our target and the numbers we were needing... He was very excited about it and basically told us that he himself could meet the need as far as attendance goes. That was exciting for us, because… there's the work that we don't have to do that somebody else can get the people for us."* – Southern Host

**Theme 2: Within the "inner setting," hosts leveraged their existing community "relational connections" to plan events and secure attendees.** "Relational connections" are high-quality formal and informal relationships, networks, and teams that exist across the inner setting, and all hosts (regardless of their recruitment success) reported that their relational connections were central to their recruitment strategy. This host describes how they used their vast community contacts to pinpoint the appropriate community to target for the research event:

*"Our strategy was to connect with our contacts and the eight counties that we serve, to see who would benefit most from the [research event] that we were about to provide. Our target group was rural African American communities, and so in all of our eight counties, we have a population of African Americans in rural areas. So, we reached out to all of them, and since we had previous relationships with them, they were very receptive to the information that we were going to provide in this [research event]."* – Southern Host

Similarly, this host described how they used their existing community connections to start their participant recruitment:

*"We reached out to a couple contacts that we had in the community, specifically with a couple African American churches, because we really wanted to focus on that population and we knew we already had some relationships there to begin with."* – Southern Host

Another host described leveraging their network to communicate to partners and to advertise the event:

*"I put up flyers all over the clinic and I distributed flyers through the clinic network, through the nurse managers, nurse managers talked about the event at staff huddles. I contacted all of my colleagues in the area, because I've been working as a nurse for 25 years in this area. So, I have Spanish speaking colleagues from a bunch of prior jobs. So, I emailed people. I reached out to organizations in the general area who center advanced care planning, palliative care, outreach to native Spanish speaking communities."* – Northeastern Host

**Theme 3: Spanish-speaking event hosts, in particular, were intentional in their planning to host an event that would be well received by the community, demonstrating a "culture of recipient centeredness."** In the CFIR, a

"culture of recipient centeredness" is defined as having shared values, beliefs, and norms around caring, supporting, and addressing the needs of recipients (who in this context, would be participants of the community research event).The Spanish-speaking event hosts in this sample (n=8) were more attuned to the underrepresented aspect and cultural sensitivity of the community where they were hosting their event. One Spanish-speaking event host was very intentional about using appropriate language that would be well received by the community.

*"I was so nervous going into it, because even though I speak Spanish every day, all day with my patients, the Spanish that I use as myself being a non-native Spanish speaker I don't ever present in front of a group. I'm only talking to an individual in front of just the two of us... I did a lot of practicing where I wrote up what I wanted to say and I practiced it and I checked my Spanish to make sure that it was okay."* – Southern Host

Similarly, another host cited:

*"This was tremendous because we actually were able to focus on a completely Spanish speaking community, and we hadn't done that [before]. And I know our system was very, very proud of that. I know the people that are of Hispanic or Latino origin were elated that we did not overlook that community. And it's a way of us really reaching the diverse communities. And so I'm very, very happy with being able to pull that off."* – Northeastern Host

This contrasted with non-Spanish speaking event hosts, whose focus was more on the topic of ACP and less about being culturally sensitive. This host describes their overall goal for the event:

*"But the goal is to empower people when the time comes, and that's why we want them to complete an advanced directive. And to us personally, it does not matter what they choose in their directive because it's what they want. But we just want them to be empowered with completing a directive."* – Midwestern Host

**Theme 4: Hosts' willingness to plan community research events stemmed from the research's dual "mission alignment" with both their organizational and personal interests in promoting ACP.** For many hosts, hosting an event aligned with their organization's goals to promote ACP in the community. In the CFIR, "mission alignment" means implementing and delivering the innovation is aligned with the overarching commitment, purpose or goals of the "inner setting". Many hosts believed hosting an event aligned with their organization's community outreach goals and would promote advance directive completion in their community, as cited by this host:

*"So, the reason to participate in those is obviously, like I said, there's a little bit of selfish that it makes our tasks easier. Individuals come into our setting with an understanding of what an advanced care directive is."* – Western Host

Hosts also discussed their personal passion for the topic of ACP including personal shared experiences as hospice nurses or healthcare workers. These hosts drew from their lived experiences helping caregivers navigate difficult decisions when end-of-life conversations had not taken place, as cited by this host:

*"Being part of those conversations on a regular basis, you come to realize how unprepared we all are and how unnecessarily hard everything is. If we can get to the root of the problem before it happens, before it becomes a problem, I feel like we all collectively thought, let's make this support for before it's needed, so that during these hard times they can focus on what's more important instead of what they should have perhaps thought about before."* – Western Host

**Theme 5: "Access to knowledge and information" provided by the host training portal was critical to preparing for the research event in their community.** In the CFIR, "access to knowledge and information" refers to guidance or training for implementing the innovation. Hosts overwhelmingly expressed appreciation for and the usefulness of the host portal that was provided by the study team. One host cited:

*"Well, the most successful part of the event was all of the information and the material that you all provided prior to [the event] that helped me prepare. That event planning guide was super helpful because it allowed me just to go through the checklist and make sure I had everything."* – Southern Host

Similarly, another host noted:

*"You know, I have to say, this was my first experience doing something like this, and I thought it was so organized. The website is so user-friendly. There's so many great tools on it."* – Western Host

Another host noted how the trainings and materials allowed them to feel confident in planning an event:

*"[I] appreciated [the] thoroughness of host portal: Project Talk does all of the materials and kind of marketing prep for you. [They] gave me the PowerPoint, they gave me the flyers, they gave me the press releases, all of it right there. And you just have to kind of plug in where you're going to meet, and I think it makes us feel more reputable and it feels good."* – Southern Host

## Discussion

When implementing public health initiatives in underserved communities, effective community engagement is critical [10]. By mapping the PTT onto the CFIR domains, we identified three key features of our CBDM that may be useful for clinicians or researchers seeking to implement other interventions in underserved communities.

First, mapping helped to reveal how our model's success featured a 'push' of resources from the outer setting to hosts who 'pulled' resources inwards to deliver them to the individuals in the inner setting. This "push and pull" of resources helped host organizations implement an evidence-based ACP intervention without needing to develop or design materials on their own. The portal and project infrastructure allow host organizations that may not have otherwise had the capacity to implement an ACP event in their community to now do so with relative ease while maintaining fidelity to the interventional approach.

Second, the CBDM places hosts at the very center of the model. Mapping the project highlighted how hosts span both the outer and inner settings and serve as a key connection between the two. Hosts, who are familiar with their community norms and culture, can begin to overcome barriers related to healthcare distrust and lack of engagement through their involvement as 'the bridge.' While 'dropping' tools into the inner setting without a trusted host facilitator is unlikely to lead to successful uptake in communities that are distrustful of the health system, having respected hosts facilitate events demonstrates respect for, and connection to, community members. Interventions grounded in caring and collaborative relationships are generally more successful than "top-down" approaches to disseminating public health information. Using hosts to span both the inner and outer settings can help to reach individuals who are often disenfranchised by the health care system.

Finally, the trial's national setting revealed the importance of flexibility in the implementation strategy, which proved critical for ensuring a successful and well-attended event. Variability in implementation needs across the outer setting's different regions became apparent and complex. When these challenges arose, hosts translated this variability into tailored solutions within their own communities, serving again as a pathway connecting the outer and inner settings.

Building on these findings from our mapping, the qualitative themes from the thematic analysis provide a closer look at how hosts navigated implementation processes and leveraged inner setting dynamics to foster community engagement and overcome barriers. The themes that centered on the "implementation process" and "inner setting" domains allowed us to better understand the intricacy of the host role and how community organizations host successful research events utilizing this model. During the PTT's "implementation process," community hosts utilized a variety of strategies to plan events and secure attendees. CFIR constructs most frequently identified from the interviews included "teaming" (joining together to coordinate and collaborate on independent tasks to implement the innovation) and "engaging" (attracting and encouraging participation in the innovation). A significant finding was the success of the CBDM in rural areas, which have traditionally been considered challenging for recruitment. All rural hosts in our study exceeded their target participant goals and described employing a multi-faceted recruitment approach. This success challenges the notion of rural participants being "hard to reach" and suggests that the CBDM (and leveraging trusted community sources) can be a powerful tool for overcoming the outsider status often attributed to researchers in these communities [32].

Within the "inner setting," qualitative themes centered on "relational connections" (the relationships, networks, and teams within and across the inner setting), "culture of recipient centeredness" (sharing values, beliefs, and norms around addressing the needs of recipients), "mission alignment" (the innovation implemented aligns with the purpose or goals of the inner setting), and "access to knowledge" (accessible training to implement and deliver the innovation). The driving force of the CBDM is "relational connections" and the communication that occurs within the inner setting to plan for events. Community hosts in the PTT relied heavily on their existing networks to plan, advertise, and host the research event, as the study team explores community connectedness and history of event planning during the interview process. A key factor in the CBDM's success was the alignment of the research topic with the host organization's mission and goals. Hosts in the PTT reported being eager to bring ACP-related interventions to their communities because they understood and appreciated the value of ACP. This finding has implications for adapting the CBDM to other health-promoting disciplines, suggesting that mission alignment should be a primary consideration when selecting community partners for research studies and programs.

Community hosts who held Spanish-speaking events were more focused on how the elements of their community event would be received by the Hispanic/Latino community. These hosts worked to ensure Spanish translation would be well received by their community's specific Spanish dialect (e.g., Mexican Spanish, Puerto Rican Spanish) and recognized the importance of the event they were hosting, given that Spanish-speaking communities are often overlooked in research. In contrast, such attention to community culture was less qualitatively evident from hosts in Black and White communities, who seemed more focused on ACP promotion and made little to no mention of needing to align particular cultural values or practices.

Preparing hosts to be successful in planning and conducting interventions in their community (by providing appropriate trainings, tools, and resources) is paramount to the successful implementation of the CBDM. Hosts in the PTT found the intervention training, research training, and recruitment materials incredibly helpful when organizing and planning their event. The training modules and advertisement materials provided via the host portal within the "inner setting" left hosts well equipped to organize the research event in their community.

## Broader implications and limitations

The findings from this implementation evaluation provide valuable lessons for researchers and practitioners seeking to use community-based models to address health disparities. The CBDM's reliance on trusted community hosts to bridge inner and outer settings, combined with its flexibility in adapting to local contexts, underscores its potential for scalability and sustainability. These findings highlight the importance of cultural competence and mission alignment in community-engaged research.

Future studies should explore the adaptability of the CBDM to other health interventions beyond ACP, especially those targeting underserved populations. By continuing to refine this model and address barriers, researchers can enhance its generalizability and effectiveness in improving health equity across diverse settings.

Limitations of the study include limited generalizability of the CBDM, as the model has only been utilized for ACP-related events. Because the PTT does not collect demographic data on the hosts themselves, we are unable to explore the influence of contextual factors, such as age, race, and gender of the host, and whether there were qualitative differences between implementation strategies and outcomes. Additionally, we did not collect data related to resource expenditures of hosts, limiting our understanding of how resource allocation might relate to event success.

Despite these limitations, the qualitative data collected were from a large-scale, national research trial where we were able to interview hosts from a diverse sample of community contexts (e.g., rural, urban, Spanish-speaking). These data are, to our knowledge, the largest examination of implementation of ACP research in communities to date. The host experiences shared help to demonstrate the barriers and facilitators to implementing community-engaged research using a CBDM. Our work highlights how implementation science can be used as a tool to better understand how methods of community-engaged research are actualized in community settings.

## Conclusions

This study demonstrates the utility of the CBDM in implementing advance care planning interventions within underserved communities. By leveraging trusted community hosts, the model effectively bridges the outer and inner settings, addressing barriers related to healthcare distrust and promoting engagement. The flexibility of the CBDM allows for tailored approaches to meet the specific needs of diverse populations, ensuring cultural relevance and fostering trust.

Our findings underscore the importance of mission alignment, relational connections, and access to training and resources in successful implementation. These elements not only support intervention fidelity but also promote sustainability by aligning the intervention with the goals and values of host organizations.

The CBDM holds promise as a scalable and adaptable model for implementing health interventions in underserved communities. Future research should expand the application of this model beyond ACP to other public health priorities, ensuring its relevance and effectiveness in diverse contexts. By addressing barriers and leveraging community strengths, the CBDM offers a pathway toward reducing health disparities and promoting equity in health outcomes.

## Acknowledgments

The study authors thank the Nick Jehlen at Common Practice, LLC (creator of the *Hello* game), Kate DeBartolo and Patty Webster, MPH and the team at *The Conversation Project* (an initiative of the Institute for Healthcare Improvement) and the team at *TableTopics* for their support of this project. We also thank Kenneth Doka, PhD, MDiv, and our safety monitoring committee members Jennifer McCormick, PhD, MPP and David Mauger, PhD and the John and Wauna Foundation for their support of the pilot work that led to this trial. Finally, we sincerely thank for our host and research participants for their engagement and participation in this project.

### The Project Talk Trial Collaborative Working Group Members

Lauren J Van Scoy, MD[1-3], Samaa Ahmed[1], BS, Cindy Bramble[4], Vernon M. Chinchilli,PhD[3], Heather Costigan,BS[1], Anna Costello,BS[4],Lindsey Currin, MSc[4], Zuiry Ghatan,BS[4,] Denise Grant, MA[4], Michael J. Green, MD[1,2], Christopher Hollenbeak, PhD[3], Kylee Kimbel,BS[1], Sara Marlin,MS[3], Terrance Murphy, PhD[3], Angela Novas,MSN,RN,CRNP[4], Allison M. Scott, PhD[5], Erika VanDyke, MPH[1], Emily Wasserman, MAS[3], Amy Tucci, BS[4]

## Author contributions

**Conceptualization:** William Calo, Benjamin Levi, Amy Tucci, Lauren Jodi Van Scoy.

**Formal analysis:** Erika VanDyke, Lauren Jodi Van Scoy.

**Writing – original draft:** Erika VanDyke, Lauren Jodi Van Scoy.

**Writing – review & editing:** Erika VanDyke, William Calo, Benjamin Levi, Amy Tucci, Lauren Jodi Van Scoy.

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
