## [Decision Letter · Decision Letter 0]

20 Jun 2025

Dear Dr. VanDyke,

Thank you for submitting your manuscript to PLOS ONE. After careful consideration, we feel that it has merit but does not fully meet PLOS ONE’s publication criteria as it currently stands. Therefore, we invite you to submit a revised version of the manuscript that addresses the points raised during the review process.

We look forward to receiving your revised manuscript.

Kind regards,

Ernesto Iadanza, PhD

Academic Editor

PLOS ONE

Journal Requirements:

“The authors declare that the research was conducted in the absence of any commercial or financial relationships that could be construed as a potential conflict of interest. Dr. Van Scoy is an unpaid scientific advisor for Common Practice, LLC.”

3. One of the noted authors is a group or consortium: The Project Talk Trial Collaborative Working Group Members

In addition to naming the author group, please list the individual authors and affiliations within this group in the acknowledgments section of your manuscript. Please also indicate clearly a lead author for this group along with a contact email address.

4. Please ensure that you refer to Figure 1 in your text as, if accepted, production will need this reference to link the reader to the figure.

5. We note that Figures 1 and 2 in your submission contain map images which may be copyrighted. All PLOS content is published under the Creative Commons Attribution License (CC BY 4.0), which means that the manuscript, images, and Supporting Information files will be freely available online, and any third party is permitted to access, download, copy, distribute, and use these materials in any way, even commercially, with proper attribution. For these reasons, we cannot publish previously copyrighted maps or satellite images created using proprietary data, such as Google software (Google Maps, Street View, and Earth). For more information, see our copyright guidelines: http://journals.plos.org/plosone/s/licenses-and-copyright.

1) You may seek permission from the original copyright holder of Figures 1 and 2 to publish the content specifically under the CC BY 4.0 license.

2) If you are unable to obtain permission from the original copyright holder to publish these figures under the CC BY 4.0 license or if the copyright holder’s requirements are incompatible with the CC BY 4.0 license, please either i) remove the figure or ii) supply a replacement figure that complies with the CC BY 4.0 license. Please check copyright information on all replacement figures and update the figure caption with source information.

If applicable, please specify in the figure caption text when a figure is similar but not identical to the original image and is therefore for illustrative purposes only.

7. We note that there is identifying data in the Supporting Information file < UTF-8PTT approvals.pdf>. Due to the inclusion of these potentially identifying data, we have removed this file from your file inventory. Prior to sharing human research participant data, authors should consult with an ethics committee to ensure data are shared in accordance with participant consent and all applicable local laws.

-Location data

Please remove or anonymize all personal information ensure that the data shared are in accordance with participant consent, and re-upload a fully anonymized data set. Please note that spreadsheet columns with personal information must be removed and not hidden as all hidden columns will appear in the published file.

Reviewers' comments:

Reviewer's Responses to Questions

**Comments to the Author**

1. Is the manuscript technically sound, and do the data support the conclusions?

Reviewer #1: Yes

Reviewer #2: Partly

Reviewer #3: Yes

2. Has the statistical analysis been performed appropriately and rigorously?

Reviewer #1: N/A

Reviewer #2: Yes

Reviewer #3: Yes

3. Have the authors made all data underlying the findings in their manuscript fully available?

Reviewer #1: Yes

Reviewer #2: Yes

Reviewer #3: Yes

4. Is the manuscript presented in an intelligible fashion and written in standard English?

Reviewer #1: Yes

Reviewer #2: Yes

Reviewer #3: Yes

Reviewer #1: Introduction:

1. In the second paragraph, the author has used stated 'burdensome and unwanted treatments', I believe this wordings is conflicting and would suggest authors to reword the statement.

Methodology:

1. Under CFIR mapping, two authors reviewed the transcripts. What was the procedure followed in instances of difference of opinion?

2. The author conducted 24 interviews. Why only 24? Is there a reasoning behind sample size fixed at 24 host participants?

3. Were there any non English audio files? If yes, how were they translated and back translated before transcribing?

4. Any particular reason why both analyst didn't code similar number of transcripts?

Results:

1. Throughout the methodology, the authors have stated 24 host participants. However, under host characteristics section, the manuscripts reads as 26. Kindly clarify.

2. Same in table 1.

Reviewer #2: The manuscript presents a valuable and timely effort to evaluate a model of community-engaged research in advance care planning (ACP) using the Consolidated Framework for Implementation Research (CFIR). The integration of CFIR into this context is conceptually sound and offers a novel application of implementation science to community-based health initiatives. However, several points merit clarification or further elaboration.

Firstly, the rationale for selecting CFIR over other frameworks (such as RE-AIM or the Theoretical Domains Framework) needs to be more explicitly stated. While CFIR is comprehensive, its application to community-engaged research—especially in culturally sensitive domains like ACP—requires adaptation and justification, particularly for constructs that may not fully resonate with community dynamics.

The manuscript could benefit from a more detailed discussion on how CFIR constructs were operationalized during data collection and analysis. For example, were interview guides or coding frameworks co-developed with community partners? Such steps would support the authenticity of community engagement and demonstrate methodological rigor. Additionally, the process for ensuring inter-coder reliability when mapping qualitative data to CFIR constructs is unclear. More transparency here would bolster confidence in the study’s analytical rigor.

In terms of findings, the manuscript thoughtfully identifies relevant CFIR domains such as “Inner Setting” and “Characteristics of Individuals,” but it is less clear how these domains interacted or evolved over time within the implementation process. A visual depiction or matrix linking specific constructs to key findings would enhance clarity and utility for readers and future researchers.

The manuscript also briefly touches on adaptability and sustainability—two crucial components of implementation success—but does not delve deeply into how these were influenced by the broader socio-political context or varying community characteristics. Exploring these nuances could offer richer insight into transferability, especially across underserved or diverse populations.

A strength of the manuscript is its integration of community voices throughout the evaluation process. However, it would be helpful to understand whether community stakeholders were involved in selecting or interpreting CFIR constructs, and how their perspectives were integrated into the final framework application. This would further demonstrate the authenticity of the “community-engaged” approach.

From an ethics and publication standpoint, there are no immediate concerns regarding dual publication or research misconduct. However, the authors should clarify if this model has been described in any prior evaluations or pilot studies. If so, appropriate cross-referencing or citation should be provided to ensure transparency and avoid redundancy.

Overall, this manuscript makes a meaningful contribution to the intersection of implementation science and community-engaged research. With greater clarity in the methodological application of CFIR, enhanced attention to context, and deeper integration of stakeholder perspectives, the study could serve as a strong model for others aiming to implement and evaluate ACP initiatives in diverse community settings.

Reviewer #3: Thank you for the opportunity to review this important and timely study. I have a few minor revisions and clarifications to recommend:

Qualitative Analysis Methods:

In the methods section, under Qualitative Analysis, please include the company name and country of origin after mentioning MAXQDA (v. 24). For example: MAXQDA (v. 24; VERBI GmbH, Berlin, Germany).

Data Consistency:

Please ensure consistency between the narrative text and Table 1 by updating the following:

In the main text, revise “Black communities (39%)” to “Black communities (38.5%)” to match Table 1.

Revise the regional distribution text to reflect one decimal place, consistent with Table 1 formatting:

“Hosts from a total of 18 states were included in the sample from across the Northeast (19.2%), Midwest (15.4%), South (42.3%), and West (23.1%).”

Table 1 Corrections:

Update the “Urban Black” entry from 38.4% to 38.5%.

Revise “West” from 23.2% to 23.1%.

Figure 1 Quality:

The image quality of Figure 1 is currently suboptimal. Please consider uploading a higher-resolution version to improve clarity for readers.

**Do you want your identity to be public for this peer review?** For information about this choice, including consent withdrawal, please see our For information about this choice, including consent withdrawal, please see our Privacy Policy .

Reviewer #1: **Yes:** Miyola Cia FernandesMiyola Cia Fernandes

Reviewer #2: **Yes:** SUBIA EKRAMSUBIA EKRAM

Reviewer #3: **Yes:** Ala ElhelaliAla Elhelali

While revising your submission, please upload your figure files to the Preflight Analysis and Conversion Engine (PACE) digital diagnostic tool, https://pacev2.apexcovantage.com/ . PACE helps ensure that figures meet PLOS requirements. To use PACE, you must first register as a user. Registration is free. Then, login and navigate to the UPLOAD tab, where you will find detailed instructions on how to use the tool. If you encounter any issues or have any questions when using PACE, please email PLOS at . PACE helps ensure that figures meet PLOS requirements. To use PACE, you must first register as a user. Registration is free. Then, login and navigate to the UPLOAD tab, where you will find detailed instructions on how to use the tool. If you encounter any issues or have any questions when using PACE, please email PLOS at figures@plos.org . Please note that Supporting Information files do not need this step.. Please note that Supporting Information files do not need this step.

---

## [Author Response · Author response to Decision Letter 1]

23 Dec 2025

Please see attached response letter.

---

## [Editor Report · Decision Letter 1]

4 Feb 2026

Using the Consolidated Framework for Implementation Research to evaluate a model of community-engaged research in advance care planning

PONE-D-25-11461R1

Dear Dr. Van Scoy,

We’re pleased to inform you that your manuscript has been judged scientifically suitable for publication and will be formally accepted for publication once it meets all outstanding technical requirements.

Kind regards,

Giovanni Ottoboni, Psy, PhD

Academic Editor

PLOS One
---

## [Editor Report · Acceptance letter]

PONE-D-25-11461R1

PLOS One

Dear Dr. Van Scoy,

I'm pleased to inform you that your manuscript has been deemed suitable for publication in PLOS One. Congratulations! Your manuscript is now being handed over to our production team.

Kind regards,

on behalf of

Professor Giovanni Ottoboni

Academic Editor

PLOS One